# The Effect of Protein Nutritional Support on Inflammatory Bowel Disease and Its Potential Mechanisms

**DOI:** 10.3390/nu16142302

**Published:** 2024-07-17

**Authors:** Qingye Li, Jing Wang

**Affiliations:** Institute of Food and Nutrition Development, Ministry of Agriculture and Rural Affairs, Beijing 100081, China; liqingye628@163.com

**Keywords:** protein nutritional support, inflammatory bowel disease, mechanisms of action, clinical application, IBD complications

## Abstract

Inflammatory bowel disease (IBD), a complex chronic inflammatory bowel disorder that includes Crohn’s disease (CD) and Ulcerative Colitis (UC), has become a globally increasing health concern. Nutrition, as an important factor influencing the occurrence and development of IBD, has attracted more and more attention. As the most important nutrient, protein can not only provide energy and nutrition required by patients, but also help repair damaged intestinal tissue, enhance immunity, and thus alleviate inflammation. Numerous studies have shown that protein nutritional support plays a significant role in the treatment and remission of IBD. This article presents a comprehensive review of the pathogenesis of IBD and analyzes and summarizes the potential mechanisms of protein nutritional support in IBD. Additionally, it provides an overview of the clinical effects of protein nutritional support in IBD and its impact on clinical complications. Research findings reveal that protein nutritional support demonstrates significant benefits in improving clinical symptoms, reducing the risk of complications, and improving quality of life in IBD patients. Therefore, protein nutritional support is expected to provide a new approach for the treatment of IBD.

## 1. Introduction

Inflammatory bowel disease (IBD) is a group of diseases characterized by chronic inflammation of the bowel, mainly divided into Crohn’s disease (CD) and Ulcerative Colitis (UC) [1]. IBD manifests as chronic inflammation of the intestinal mucosa, accompanied by abdominal pain, diarrhea, anemia, and other symptoms, which seriously affects the quality of life of patients [2]. Globally, the prevalence of IBD is on the rise, making it one of the focal points in public health. As of 2019, there were 4.89 million cases of IBD worldwide, with an estimated 400,000 new cases of IBD in that year alone [3,4]. Developed countries have consistently been the primary regions with high prevalence of IBD, especially North America, Europe, and Australia [3]. However, in recent years, developing countries have also experienced a rise in IBD prevalence, a trend that has attracted considerable attention from researchers because it may be related to changes in lifestyle, dietary patterns, and environmental factors [5].

The pathogenesis of IBD involves multiple factors such as genetics, immunology, environment, and gut microbiota [6]. Early studies primarily focused on exploring the association of genes and immune system abnormalities with IBD. However, recent studies suggest that environmental factors (such as diet and lifestyle) and changes in the microbial community are also considered closely related to the occurrence of IBD [7,8]. Clinically, the treatment of IBD primarily focuses on medication, with surgery as a necessary option in certain cases. Anti-inflammatory drugs such as 5-aminosalicylic acid and glucocorticoids are used to alleviate acute symptoms [9]. Immunomodulatory drugs such as azathioprine can be employed to maintain long-term remission [10,11,12]. Biologics, including anti-TNF drugs and other drugs targeting immune pathways, have become crucial therapeutic methods in IBD treatment by modulating the immune response and targeting inflammatory factors [13,14,15]. Surgical treatment is primarily utilized to address severe complications or cases where medical therapy is ineffective. Although drug treatment can effectively control disease symptoms and improve patients’ quality of life, it also presents a series of problems such as significant side effects, drug resistance, high costs, drug dependence, and suppression of the immune system [16,17].

In recent years, people have gradually realized the importance of nutrition to health, and nutritional intervention plays a crucial role in disease prevention, treatment, and postoperative recovery [18,19,20,21]. With protein as the most essential nutritional component, studies have found that high-quality animal and plant proteins in the diet, especially whey protein and soy protein, have the role of regulating the body’s immune system, reducing inflammatory reactions, antioxidant properties, and so on [22,23,24,25]. Animal and clinical studies have revealed that high-quality protein plays a significant role in the treatment of IBD [26,27,28,29]. Protein nutritional support in IBD patients can improve the nutritional status of the body, reduce inflammatory indicators, mitigate intestinal mucosal damage, and improve clinical outcome indicators [30,31,32]. This article reviews the pathogenesis of IBD, the impact of protein nutritional support on IBD, and its potential mechanisms. Additionally, this article comprehensively analyzes the application of protein nutritional support in clinical practice and its impact on IBD complications. We hope that this review will provide new theoretical perspectives and practical references for the subsequent application of protein nutritional support in the clinical treatment of IBD.

## 2. Pathogenesis

The pathogenesis of IBD is a complex process that has not been fully elucidated despite extensive research. Currently, it is known that multiple factors, including genetics, immunity, environment, and intestinal microbiota all play indispensable roles in IBD, and they influence each other to constitute the pathogenesis of this disease [33].

### 2.1. Genetics

Genetic factors play a significant role in the pathogenesis of IBD. One important phenomenon observed in IBD is the significantly higher incidence rate among first-degree relatives of IBD patients compared to the general population [34,35,36,37,38,39]. Through analytical methods such as genome-wide association studies (GWASs) and next-generation sequencing, researchers have identified over 240 non-overlapping IBD genetic risk loci, with approximately 30 gene loci associated with both CD and UC [40,41,42]. Analysis shows these genes are involved in pathways crucial for intestinal homeostasis [43,44,45]. However, the heritability estimated through GWASs is significantly lower than those obtained from twin studies, and this discrepancy can be attributed to the complex role of epigenetics [43,46]. Epigenetics has a profound impact on gene phenotypes through multiple factors such as the environment and gut microbiota, thereby participating in the pathogenesis of IBD. Data from the UK Biobank indicate that, among adult IBD patients with a high genetic risk, maintaining a healthy lifestyle can potentially reduce the risk by half [47]. By delving deeper into the genetic background and pathogenesis of IBD, we can actively and effectively prevent and manage IBD, thereby providing better health management for patients.

### 2.2. Immune Response

Some studies have shown that innate and adaptive immune responses play a crucial role in the development of IBD. Innate immunity is the first line of defense, providing a rapid, non-specific immune response, while the adaptive immune system enhances defenses through specific responses, such as T and B cell activation, producing targeted immune reactions against specific pathogens or antigens [48]. Under normal conditions, the immune system can recognize and distinguish between its own tissues and foreign pathogens, maintaining immune tolerance. However, when the intestinal mucosal barrier is damaged, the intestinal flora is disturbed, immune regulatory cells are abnormal, or autophagy is dysfunctional, it can lead to abnormal immune responses and trigger the development of intestinal inflammation [49]. The impairment of the intestinal mucosal barrier is a significant contributing factor to the occurrence of IBD [50,51]. When the intestinal mucosal barrier is damaged, the increased mucosal permeability exposes the intestinal tissues to a large number of antigens, leading to an overreaction and misidentification by the intestinal immune system, activating macrophages and lymphocytes to release a significant amount of cytokines and inflammatory mediators, ultimately leading to tissue damage through the progressive amplification of inflammatory immune responses [52].

### 2.3. Intestinal Microbiota

Intestinal microbiota are often influenced by environmental factors such as diet, antibiotics, mental factors, and trauma [53]. Changes in these factors may disrupt the balance of intestinal microbiota, leading to damage to the intestinal mucosal barrier, triggering the release of inflammatory mediators, and resulting in abnormal activation of immune cells, thereby exacerbating the development of intestinal inflammation [54,55]. Animal studies have shown that the intestinal microbiota is an indispensable factor driving the pathogenesis of IBD [53]. However, it is challenging to establish a clear causal relationship between the intestinal microbiota and IBD in humans, as a single infection is unlikely to cause or trigger IBD, but the intestinal microbiota clearly promotes the development of IBD [56,57]. Compared with healthy individuals, a significant decrease in gut microbial diversity was observed in IBD patients, with a decrease in bacteria with anti-inflammatory effects and an increase in bacteria with pro-inflammatory effects [58,59,60,61,62]. In addition, an increase in the number of mucus-degrading bacteria was also observed in IBD patients, leading to the erosion of the intestinal mucus barrier, prompting more pathogens and microbiota products (such as toxins and lipopolysaccharides) to enter the intestinal epithelium and trigger intestinal inflammation [63,64]. It is noteworthy that the number of certain bacteria capable of producing short-chain fatty acids, such as *Faecalibacterium prausnitzii*, has also been shown to decrease in IBD patients, and these short-chain fatty acids play crucial roles in regulating cytokine release, modulating intestinal immunity, and providing energy to intestinal epithelial cells [65]. In addition to bacterial changes, there were also significant changes in the fungal community within the gut microbiota of IBD patients, including an increase in fungal diversity and fungal dysbiosis [66]. A higher fungal diversity was positively correlated with the severity of inflammation and the levels of inflammatory markers such as TNF-α and IL-10 [67]. In summary, changes in the intestinal microbiota are closely linked to the pathogenesis of IBD. A deeper understanding of the relationship between the intestinal microbiota and IBD will help us find more effective prevention and treatment strategies to maintain intestinal health and alleviate patient suffering.

### 2.4. Environment

The incidence of IBD in developing countries has significantly increased in the past decade, and it has gradually become one of the common diseases of the digestive system from the past state of being a rare disease [4]. This trend indicates that environmental factors have a significant impact on the onset of IBD, especially for genetically susceptible individuals, where environmental changes may be one of the important factors triggering intestinal inflammation [8,68]. Among these, some specific environmental factors have been confirmed to be related to the occurrence of IBD. For example, ultra-processed foods, smoking, antibiotic exposure, appendectomy, tonsillectomy, oral contraceptives, environment pollution, and infant feeding patterns can all increase the risk of IBD occurrence [69,70,71,72,73]. Diet is a significant environmental factor influencing the occurrence and severity of IBD. The rise in global IBD incidence can be partially attributed to the shift towards Western dietary patterns and excessive consumption of processed foods in newly industrialized countries [33,74]. The modern Western diet is characterized by excessive intake of animal-derived proteins, refined grains, hydrogenated fats, and processed foods, with a decrease in the consumption of fruits and vegetables [33]. Changes in dietary patterns lead to alterations in the quality and proportion of nutrition, resulting in intestinal dysbiosis and subsequently triggering intestinal inflammation. Clinical trials have shown that IBD patients who adopt an exclusionary diet (reducing refined sugar, saturated fats, emulsifiers, red meat, and ultra-processed meat) can better maintain clinical remission and improve intestinal inflammatory responses [75]. Smoking is one of the most extensively studied factors in the pathogenesis of IBD. Clinically, smoking has been proven to be harmful to patients with CD, while it appears to have a protective effect on those with UC [43]. The use of drugs, especially antibiotics, is also associated with an increased risk of IBD [76,77]. This correlation is often attributed to the long-term use of antibiotics, which can cause changes in bacterial flora, increasing the risk of infection by potential pathogenic bacteria and consequently leading to intestinal inflammation [78]. Long-term exposure to certain pesticides, heavy metals, or other harmful chemicals can disrupt intestinal barrier function, leading to the occurrence of intestinal inflammation [79,80]. At the same time, environmental issues such as air pollution and water pollution also have negative impacts on intestinal health [81,82]. Overall, the impact of environmental factors on IBD is multifaceted, including lifestyle, dietary habits, and exposure to certain environments. However, the specific mechanisms of these environmental factors are not fully understood at present and require further research for deeper exploration. Moreover, modifying environmental factors such as lifestyle and dietary habits may help reduce the risk of developing IBD or improve disease symptoms.

## 3. Mechanisms of Protein Nutritional Support in Alleviating IBD

Protein nutritional support has gradually shown its unique advantages in the treatment of IBD. Through animal experiments and clinical studies, we found that protein not only provides the necessary energy and nutrition for the human body as the cornerstone of life activities, but also participates in the remission process of IBD through a series of complex mechanisms (Table 1). Analyzing its possible mechanisms, specifically, protein nutritional support can effectively promote the secretion of intestinal mucin, enhance intestinal barrier function, and thereby resist the invasion of harmful substances; regulate the expression of intestinal tight junction proteins, maintain the stability of intestinal structure, and reduce the penetration of inflammatory substances; regulate intestinal microbiota, maintain intestinal microecological balance, and promotes intestinal health. In addition, the protein also affects the activity of key signaling pathways, such as NF-κB and Keap1/Nrf2/HO-1, further alleviating IBD symptoms by inhibiting inflammatory responses and enhancing antioxidant capacity (Figure 1). Therefore, delving deeper into the mechanisms of how protein nutritional support alleviates IBD is of significant importance for optimizing nutrition treatment plans for IBD patients and improving treatment outcomes.

### 3.1. Promote Mucin Secretion

Mucins are huge and highly negatively charged glycoprotein secreted by goblet cells of intestinal epithelium [95]. They are an important part of the intestinal mucus layer, which acts as a barrier between bacteria and the intestinal epithelium [96]. The mucins can be classified into two types: gel-forming mucins and transmembrane mucins [97]. At present, 20 different mucins have been identified [98]. Mucin 2 (MUC2), known as gel-forming mucin, is the main component of the intestinal mucus, and forms the skeleton of the intestinal mucus layer through the connection of disulfide bonds [99,100]. Mucin production is critical as reports suggested an increased risk of colitis and inflammation-induced colorectal cancer in MUC2 deficient mice [101]. Clinical studies have found that the expression of MUC2 in the colon of patients with UC is significantly reduced, and the expression of MUC2 was negatively correlated with the severity of UC [102]. However, MUC2 expression in Crohn’s patients fluctuates irregularly with clinical activity of the disease [103,104].

Regarding the relationship between dietary protein and mucin, the initial study found that milk protein hydrolysates can induce a large amount of mucin release from intestinal goblet cells by activating the enteric nervous system and opioid receptors [105,106]. The presence of opioid receptors on intestinal cells suggests that food-derived peptides with the structure of opioid receptor agonists, which can be produced in the intestinal lumen during gastrointestinal digestion, might regulate mucin production via a direct action on epithelial goblet cells [107]. Studies in animal experimental models of colitis have also found that whey protein and soy protein as dietary protein sources can increase the secretion of colonic MUC2 to reduce intestinal mucosal damage [84,85]. However, some researchers support the view that the protective effect of whey protein against colitis induced in rats is due to its high levels of threonine and cysteine, which stimulate MUC2 synthesis [86,108]. Zeng et al. found that feeding soy protein isolate to weanling mice reduced intestinal MUC2 production and weakened intestinal immune function in mice, which should be take into account when using soy protein as a dietary protein source in children or young animals [109]. Therefore, it is still needed to further explore the mechanism of dietary protein regulation of MUC2.

### 3.2. Regulate Intestinal Tight Junction Proteins

Intestinal tight junction proteins are important molecules that form tight connections between intestinal cells and play a key role in maintaining the integrity and permeability of the intestinal mucosal barrier [110]. Studies show that Occludin, Claudins, and Zonula Occluden 1 (ZO-1) are the most important three types of tight junction proteins [111,112]. When the expression or distribution of intestinal tight junction proteins are abnormal, the integrity of the intestinal mucosal barrier is damaged, and the permeability is increased, which may lead to intestinal inflammation, allergy, and other problems [113,114]. All IBD patients were accompanied by intestinal mucosal barrier impairment to varying degrees. Tumor necrosis factor-α (TNF-α) is a key inflammatory factor that leads to impaired intestinal mucosal barrier function [115]. In IBD, TNF-α increases the expression/phosphorylation of myosin light chain kinase (MLCK) by inducing NF-κB activation, which subsequently leads to the delocalization of tight junction proteins on intestinal epithelial cell membranes, ultimately resulting in increased intestinal permeability [116,117,118,119]. Many experimental studies have shown that whey protein and soy protein can significantly inhibit the expression of TNF-α in the colon of colitis animals, improve intestinal mucosal permeability, and reduce the score of colon inflammation [28,83,84,85,86,87]. Given the importance of protein, in recent years, clinical researchers have begun to consider the use of protein nutritional support in IBD. One of the clinical studies observed that a daily dose of whey protein could significantly relieve intestinal epithelial inflammation and improve intestinal permeability and intestinal mucosal morphology [30]. This evidence suggests that protein nutritional support regulates the distribution of tight junction proteins in intestinal epithelial cells by reducing TNF-α expression, thereby improving intestinal mucosal permeability.

### 3.3. Regulation of Intestinal Microbiota

High-quality protein, as one of the essential nutrients for the human body, plays a pivotal role in maintaining the balance of intestinal microbiota. In recent years, with the continuous deepening of research, more and more evidence has shown that high-quality protein has a positive impact on intestinal microbiota [120,121,122,123]. As one of the main nutritional components in food, the quality and quantity of protein intake directly affect the nutritional status of intestinal microbiota. When the human body consumes adequate high-quality protein, these proteins are digested into small molecules in the stomach and small intestine, including small peptides and amino acids, which then enter the colon to become a food source for intestinal microbiota. Intestinal microbiota utilizes these amino acids for growth and reproduction, thereby maintaining the diversity and activity of the intestinal microbiota. Studies have found that consuming appropriate amounts of high-quality protein can improve the structure of the intestinal microbiota by increasing the proportion of beneficial bacteria and reducing the number of harmful bacteria [124,125]. In addition, high-quality protein also affects the production of intestinal microbial metabolites. Intestinal microbiota produce a series of compounds, such as short-chain fatty acids, through the metabolism of amino acids. Short-chain fatty acids play important roles in maintaining intestinal mucosal barrier function, providing energy, regulating the immune response, and having anti-inflammation effects [126,127,128]. Multiple studies have shown that moderate intake of high-quality protein can promote the production of short-chain fatty acids, thereby contributing to maintaining intestinal health [27,129].

Previous discussions have addressed the significance of intestinal microbiota in the onset and progression of IBD. Given the positive impact of high-quality protein nutritional support on intestinal microbiota, an increasing number of studies have begun to focus on the application of protein nutritional support in IBD. It was found that replacing animal protein in a regular diet with a mixture of soy protein and pea protein can alter the composition of intestinal microbiota in mice with DSS-induced colitis, increasing Lactobacillaceae abundance and promoting an increase in the concentration of the metabolites glutamine and butyric acid, thereby reducing the severity of experimental IBD [28]. In another trial that replaced casein in the diet of colitis rats with whey protein, it was found that whey protein increased fecal counts of Lactobacillus and Bifidobacterium, both of which have been shown to benefit gut health [86]. A recent trial with daily supplementation of a certain amount of whey protein hydrolysate has demonstrated that it can promote the growth of beneficial bacteria norank_f_Muribaculaceae in DSS-induced mice, which utilizes intestinal mucus polysaccharides as a growth nutrient, thereby inhibiting the growth and colonization of pathogenic bacteria such as Romboutsia and Enterobacter in the intestine and improving intestinal mucosal damage [27]. Other proteins, such as quinoa protein and Alaska pollack protein have also been shown to alleviate experimental colitis by adjusting intestinal microbiota structure [97,98].

### 3.4. Regulation of Key Signaling Pathways

#### 3.4.1. NF-κB

NF-κB is an inducible transcription factor that regulates many different genes involved in regulating inflammatory processes. The NF-κB family includes five members, NF-κB1 (p50), NF-κB2 (p52), ReIA (p65), ReIB, and c-ReI. The different NF-κB members mediate the transcription of mainly inflammatory factors by binding to specific DNA sequences and forming homo- or heterodimers [130,131]. NF-κB pathways are activated by a variety of stimuli such as ligands of various pro-inflammatory cytokines receptors. NF-κB not only increases the production of pro-inflammatory cytokine, adhesion molecules, and chemokines, but also induces the survival and proliferation of inflammatory cells through the production of anti-apoptotic factors and has been shown to be the primary regulatory component of the inflammatory burden in intestinal inflammation during UC and CD [132]. The activation of NF-κB in the intestinal lumen is predominant in the intestinal macrophages and epithelial cells of the intestinal lumen mucosa, and the higher the number of cells with activated NF-κB stain, the worse the severity of the intestinal inflammation. The expression of NF-κB is accompanied by increased production of pro-inflammatory cytokines such as TNF-α, IL-1, and IL-6, which mediates intestinal mucosal tissue injury [133]. Many clinical therapeutic agents for IBD, such as glucocorticoids and 5-aminosalicylic acid, have mechanisms of action that are related to the inhibition of NF-κB [134,135,136]. Currently, inhibition of NF-κB activation during IBD is also a direction of clinical research. Animal studies have found that feeding a certain amount of whey protein or quinoa protein daily can significantly reduce the expression level of NF-κB in the colon of IBD rats [87,91,94]. Further research is needed on how high-quality proteins inhibit the NF-κB pathway in IBD.

#### 3.4.2. Keap1/Nrf2/HO-1

Kelch-like ECH-associated protein 1 (Keap1), Nuclear factor erythroid-2-related factor 2 (Nrf2), and Heme oxygenase-1 (HO-1) are important antioxidative signaling molecules that possess antioxidant properties and maintain cell integrity. The Keap1/Nrf2/HO-1 signaling pathway plays an important role in protecting intestinal cells against oxidative stress and inflammatory injury [137,138]. HO-1 as a stress-induced protein is induced by various oxidative and inflammatory signals, subsequently inducing anti-inflammatory activity. HO-1 mRNA and protein levels are up-regulated after oxidative stress and cell damage, and Nrf2 can also directly regulate HO-1 promoter activity [139]. Nevertheless, the activity of Nrf2 is precisely regulated by the negative regulator Keap1. Under the basal state, Nrf2 is sequestered in the cytoplasm by Keap1. When the cell experiences conditions of oxidative stress, electrophiles, or chemopreventive agents, Nrf2 dissociates from the Keap1-mediated inhibition state and activates the expression of genes mediated by antioxidant response element.

An experiment on colitis rates found that supplementation of whey protein could significantly increase the expression levels of Nrf2 and HO-1, and decreased the expression levels of inflammatory markers [87]. In addition, the supplementation of soy protein can also reduce the level of inflammation or oxidative stress in the body by regulating the expression levels of Nrf2 and HO-1 [140,141]. It is speculated that high-quality proteins produce peptides with antioxidant activity after digestion in vivo, which promote the transfer of Nrf2 from the cytoplasm to the nucleus by binding to Keap1 or blocking the binding of Keap1 and Nrf2 and then promote the transcription and expression of the downstream gene HO-1, thus reducing the level of inflammation in the body [142,143,144,145,146,147,148,149].

## 4. The Clinical Application of Protein Nutritional Support in IBD

With the in-depth development of medical research, the application of protein nutritional support in the clinical treatment of IBD has gradually received widespread attention (Table 2). As the basic substance of human life activities, protein is of great significance for maintaining normal physiological functions. In particular, the intake of high-quality protein can not only provide essential amino acids for the body but also strengthen the intestinal barrier function by adjusting the ecological balance of intestinal microorganisms, thereby reducing inflammation [122,150,151,152,153]. The intestinal mucosa of IBD patients is often damaged by inflammation, leading to nutrient malabsorption and affecting the overall nutritional status of patients [154]. The core of protein nutritional support is to improve the nutritional status of patients and promote the repair of intestinal mucosa by optimizing the dietary structure of patients and increasing the intake of high-quality protein. In addition, high-quality protein can also adjust the ecological balance of gut microbes, enhance intestinal barrier function, thereby reducing inflammation and relieving the symptoms of IBD (Figure 2).

In clinical practice, whey protein and soy protein, as high-quality protein sources, have shown promising applications in the treatment of IBD. In the treatment of patients with CD, the intervention of a regular diet combined with concentrated whey protein can significantly improve intestinal permeability and intestinal morphology, increase the villous crypt ratio, and reduce inflammation marker levels [30]. Another study showed that daily supplementation with whey protein or soy protein isolate can also reduce body fat percentage, increase upper arm muscle circumference, and correct upper arm muscle area in patients with CD, further improving the patient’s body composition [29]. A similar experimental study has shown that daily whey protein supplementation combined with resistance training can significantly increase skeletal muscle mass and improve hemoglobin and creatinine levels in patients with IBD [31]. In addition to oral supplementation of high-quality protein, adjusting dietary structure is also a commonly used method in IBD treatment. A study has shown that a diet rich in soy protein can effectively improve the body composition of patients with inactive and lactose-intolerant CD and increase treatment compliance [155].

In addition to whey protein and soy protein, other forms of protein nutritional support also play an important role in IBD treatment. Casein glycomacropeptide, as a milk-derived polypeptide, has a similar clinical remission rate to medications in the treatment of patients with UC and has shown good tolerance and acceptability [158]. Lactoferrin, a protein isolated from milk with anti-inflammatory and immunomodulatory effects, can significantly reduce inflammation marker levels and improve biochemical indicators such as hemoglobin and serum iron in IBD patients [160]. Furthermore, a semi-elemental diet containing hydrolyzed whey protein not only improves the nutritional status of IBD patients but also reduces the disease activity index and decreases the frequency of defecation [32]. It is worth mentioning that an amino acid-based elemental diet has also shown similar positive effects in the treatment of CD [161,162,163]. Elemental diets are not only easily absorbed by the intestines but also have the advantage of low antigenicity, which is significant for reducing intestinal burden and promoting intestinal repair. However, studies on patients with active CD have conducted a comprehensive comparison of the efficacy of different enteral formulations, and the results show that there is no statistically significant difference in treatment effect between elemental diets and non-elemental diets [164,165,166]. This indicates that the specific composition of proteins may not directly affect the overall therapeutic potential of enteral nutrition. In light of this, the European organizations for IBD and for pediatric gastroenterology and nutrition, ECCO and ESPGHAN, recommend the use of polymeric diets that are closer to daily dietary habits for patients with relatively mild disease or lower risk of relapse, in order to enhance patient acceptance and quality of life. Only in special cases where patients have milk protein allergy, a switch to elemental diets is recommended to ensure the safety and effectiveness of nutritional intake [167]. In addition, the infusion of amino acids through parenteral nutrition is also one of the important clinical pathways for protein supplementation. As a form of nutritional support, parenteral nutrition is primarily used for IBD patients who are unable to digest and absorb food normally due to severe intestinal inflammation or impaired function, as well as for patients who have not responded to other treatments [168,169]. However, long-term reliance on parenteral nutrition may lead to intestinal functional degeneration, increased risk of infection and catheter-related complications [170,171]. Therefore, parenteral nutrition is usually used as a supplement to enteral nutrition.

In summary, the application of protein nutritional support in the clinical treatment of IBD has achieved remarkable results. By adjusting the diet structure of patients, increasing the intake of high-quality protein and using specific protein interventions not only can improve the nutritional status of patients but can also reduce the symptoms of disease, control the inflammatory response, and reduce the occurrence of clinical complications. In the future, with the continuous deepening of research and technological advancements, the application of protein nutritional support in IBD treatment will become more extensive and precise, bringing benefits to more patients.

## 5. Protein Nutritional Support on IBD Complications

### 5.1. Malnutrition Associated with IBD

IBD patients are at risk of malnutrition due to reduced food intake, malabsorption, increased gastrointestinal loss, increased energy requirements due to hypercatabolism, and occasionally from drug–nutrient interactions [172]. Clinical studies have shown that IBD patients are often accompanied by varying degrees of malnutrition, and the severity of malnutrition in IBD is influenced by the activity, duration, and the magnitude of the inflammatory response [168]. Malnutrition in IBD patients should be treated appropriately, as it can worsen the prognosis, complication rates, mortality, and quality of life. When IBD patients are diagnosed, they should be screened for malnutrition and thereafter on a regular basis. Several standard nutritional risk screening tools have been used to adequately screen and assess malnutrition, including the Subjective Global Assessment (SGA), Nutritional Risk Score 2002 (NRS 2002), and Malnutrition Universal Screening Tool (MUST). In addition, there are several specific nutritional risk screening tools for IBD patients, such as Saskatchewan Inflammatory Bowel Disease–Nutrition Risk Tool (SaskIBD-NR Tool), the IBD Specific Nutrition Self-Screening Tool (IBD-NST), and the Malnutrition Inflammation Risk Tool (MIRT) for CD patients [173,174]. 

The most common type of malnutrition in IBD patients is protein-energy malnutrition. Malnutrition can lead to a decline in muscle mass and function, which can further lead to sarcopenia [175]. Studies have shown that high-quality protein supplementation can improve malnutrition in children, the elderly and perioperative populations. A randomized, double-blind trial for the treatment of childhood moderate acute malnutrition found that adding whey protein to complementary foods improved malnutrition in children [176]. Another randomized, double-blind trial in 6-to-59-month-old children suffering from severe acute malnutrition showed that children who received soy protein-based ready-to-use therapeutic foods showed similar outcomes in terms of weight gain, rate of weight gain, changes in other body measurements, and body composition compared to those who received milk-based ready-to-use therapeutic foods [177]. Older people are at high risk for malnutrition, especially those with chronic diseases. Some studies have shown that supplementing foods rich in high-quality protein can improve malnutrition in elderly people [178,179,180]. A clinical trial of allogeneic hematopoietic stem cell transplantation found that daily supplementation of soy–whey mixed protein prior to transplantation improved protein-energy malnutrition in leukemia patients compared to a natural diet group [181]. Clinical studies have found that high-quality protein supplementation plays an important role in improving malnutrition in IBD patients. Daily supplementation of whey protein or soy protein can improve the incidence of malnutrition and muscle mass in patients with IBD and colorectal cancer [29,182]. ESPEN recommends that during active IBD, protein requirements are increased, and intake should be elevated (to 1.2–1.5 g/kg/day in adults) compared to recommendations for the general population; whereas during remission, protein requirements are generally not elevated, and the recommended intake should be similar (approximately 1 g/kg/day in adults) to that for the general population [168].Therefore, IBD patients should pay attention to the intake of high-quality protein in their daily diet to prevent or improve malnutrition.

### 5.2. Sarcopenia Associated with IBD

Sarcopenia is defined as a complex syndrome characterized by a progressive, generalized decrease in muscle mass and strength. There are two types of sarcopenia: primary sarcopenia and secondary sarcopenia. Primary sarcopenia is caused by heredity or aging. Secondary sarcopenia is caused by insufficient activity, malnutrition, malignancy, congestive heart failure, chronic liver disease, chronic congestive pulmonary disease, chronic inflammation, or steroid therapy [183]. IBD, as a chronic inflammatory disease, is significantly associated with metabolic disorders and nutrient deficiencies, which subsequently increases the risk of sarcopenia. Studies reported that over one-third of IBD patients have sarcopenia and pre-sarcopenia, and nearly one-fifth have sarcopenia [184]. Sarcopenia has a negative impact on the length of hospital stay, surgical outcomes, clinical course, postoperative complication, and low bone mineral density and results in quicker biologic agent failure of IBD patients [184,185,186].

Protein supplement is an emerging treatment of sarcopenia and has attracted increasing attention. Many international working groups, such as the Asian Working Group for Sarcopenia (AWGS), the European Working Group on Sarcopenia in Older People (EWGSOP), and the Australian and New Zealand Society for Sarcopenia and Frailty Research (ANZSSFR), all recommend supplementing adequate amounts of protein, which can help prevent or even reverse sarcopenia [187,188,189]. High-quality protein is rich in essential amino acids, especially branched-chain amino acid, which can stimulate rates of muscle protein synthesis and suppress rates of muscle protein breakdown and play an important role in maintaining muscle health [190]. Clinical trials have found that supplementation of high-quality protein can improve muscle mass and reduce the incidence of sarcopenia in IBD patients. One study investigated personalized nutrition care in sarcopenic IBD patients, a protein goal of 1.2–1.5 g/kg of ideal body weight was applied in the preoperative phase (whey protein supplementation); a significant increase in muscle mass was reported after 103 days compared to baseline measurements [191]. A clinical trial for CD found that supplementation with whey protein and soy protein isolate reduced body fat and improved muscle mass [29]. Protein supplementation and resistance training are widely recommended as part of the treatment of sarcopenia. A randomized, double-blind, placebo-controlled trial in patients with IBD found that the group receiving whey protein combined with resistance training improved sarcopenia more effectively than the group receiving resistance training alone [31]. All of these findings were accompanied by reductions in clinical inflammatory indicators. It is inferred that high-quality protein may improve muscle by stimulating muscle protein synthesis and reducing muscle protein breakdown caused by inflammation, thus further reducing the incidence of sarcopenia. For patients with IBD who experience the clinical complication of sarcopenia, although there is no standardized protein intake guideline across clinical practices, it is generally advised that their daily protein intake should be at least 1.2 to 1.5 g per kilogram of body weight [192]. The specific intake should be tailored to the individual patient’s physiological condition, disease activity level, and the severity of sarcopenia. In addition to increasing protein intake, resistance training is also recommended to further augment muscle mass and strength recovery.

### 5.3. Osteoporosis Associated with IBD

Osteoporosis is a common systemic skeletal disease characterized by an imbalance in bone formation and resorption. Osteoporosis is one of the most common complications in patients with IBD, with a prevalence of 18 to 42% [193,194]. The etiology of osteoporosis in patients with IBD mainly involves several factors, such as low mineral intake, decreased vitamin D synthesis, reduced physical activity, long-term use of hormone medications, and gastrointestinal damage caused by ongoing inflammatory processes [6,195]. The cause of osteoporosis in patients with IBD is complex, which makes the prevention and treatment of IBD-related osteoporosis relatively difficult. There are several therapeutic drugs available for the treatment of IBD-related osteoporosis, such as bisphosphonates, vitamin D and calcium supplementation, calcitonin, recombinant parathyroid hormone, hormone replacement therapy, minimizing corticosteroid use, and alternative treatment [196]. However, due to the lack of sufficient clinical data, most drugs are only suitable for adults, and, in addition, they usually have some limitations such as high prices and obvious side effects [197,198]. Nutritional support as a safe and economical approach has been applied to the management of IBD and osteoporosis [199,200,201,202]. Studies have found that protein, as an important component of nutritional support, can not only improve inflammation in patients with IBD, but also contribute to the bone health of patients. Milk and dairy products are one of the main sources of high-quality protein, and limiting milk and dairy products consumption are risk factors for osteoporosis in patients with IBD. Iwona et al. found that the proportion of IBD patients who consumed milk significantly decreased after diagnosis, and the bone mineral density of IBD patients who did not drink milk was lower than that of IBD patients who drank milk [203]. Coqueiro et al. observed a moderate correlation between bone mineral density and dietary protein intake in patients with CD [157]. A study revealed that a moderately high soy protein diet in animal models of IBD mitigated the high osteoclast surface and depressed the bone formation rate, in addition to inhibiting the expression of inflammatory cytokines such as TNF-α and receptor activation of the NF-κB ligand in the gut and bone [83]. It is speculated that high-quality proteins such as milk protein and soy protein can ameliorate osteoporosis by inhibiting osteoclast-mediated bone resorption, promoting the formation and proliferation of osteoblasts, and enhancing calcium absorption [201,204]. For IBD patients at risk of osteoporosis, ensuring adequate daily intake of high-quality animal and plant proteins can not only improve their overall nutritional status but also significantly promote bone health. It is recommended to design a reasonable protein intake plan by combining the recommended intake levels outlined in the ESPEN guidelines while fully considering the individual patient’s circumstances.

## 6. Conclusions

As a disease characterized by chronic intestinal inflammation, IBD not only seriously affects the quality of life of patients, but also poses a severe challenge to global public health. Its pathogenesis is a multifaceted process involving genetics, immunology, environmental factors, and gut microbiota. Early research focused primarily on the role of genetic susceptibility and immune system abnormalities in IBD, while recent studies have shifted their focus to environmental triggers and gut microbial imbalances, indicating that these factors play a crucial role in the etiology of IBD. This shift in research perspective has opened up new avenues for exploring potential therapeutic targets and preventative measures.

As the foundation of life activities, proteins play an irreplaceable role in maintaining intestinal health, promoting intestinal mucosal repair, and regulating immune responses. This article reviews the therapeutic and ameliorative effects of protein nutritional support on IBD in recent years and analyzes its possible mechanisms of action. It is found that protein nutritional support can treat and alleviate IBD by promoting mucin secretion, regulating the distribution of intestinal tight junction proteins, regulating intestinal microbiota, and modulating two key metabolic pathways, NF-κB and Keap1/Nrf2/HO-1. The application of protein nutritional support in IBD patients aims to improve the nutritional status of patients, reduce intestinal inflammation, and thereby enhance their quality of life by optimizing protein intake. Existing studies have shown that appropriate protein nutritional support can reduce the incidence of complications in IBD patients, promote the recovery of intestinal function, and improve the prognosis of the disease to some extent (Table 2).

However, the application of protein nutritional support in the treatment of IBD is still in the exploratory stage, and many details and mechanisms remain to be further elucidated. Future research should focus more on the intrinsic link between protein and the pathogenesis of IBD, as well as the specific impacts of different protein sources and types on the intestinal health of IBD patients. In addition, personalized nutrition support programs for different populations and disease stages are also an important direction for future research. With technological advancements and deeper research, we expect the application of protein nutritional support in IBD treatment to become more precise and effective, emerging as a crucial tool for patients’ self-management and rehabilitation. Meanwhile, exploring the combined application of protein nutritional support with drug therapy, surgery, and other treatments is anticipated to offer a more comprehensive and systematic therapeutic approach for IBD patients.

## Figures and Tables

**Figure 1 nutrients-16-02302-f001:**
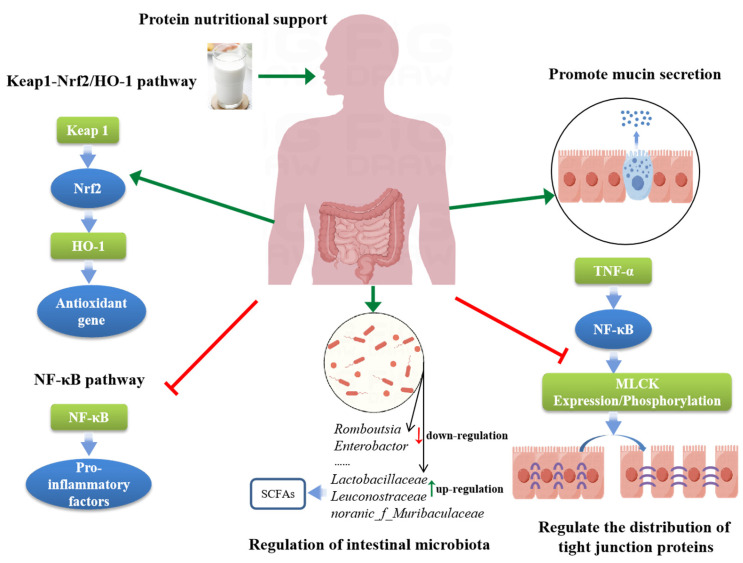
Potential mechanisms of protein nutrition improving IBD (By Figdraw, https://www.figdraw.com/static/index.html#/, 6 May 2024). Abbreviations used: HO-1, Heme oxygenase-1; Keap1, Kelch-like ECH-associated protein 1; MLCK, Myosin Light Chain Kinase; NF-κB, nuclear factor kappa-B; Nrf2, Nuclear factor erythroid-2-related factor 2; SCFAs, short-chain fatty acids; TNF-α, tumor necrosis factor-α.

**Figure 2 nutrients-16-02302-f002:**
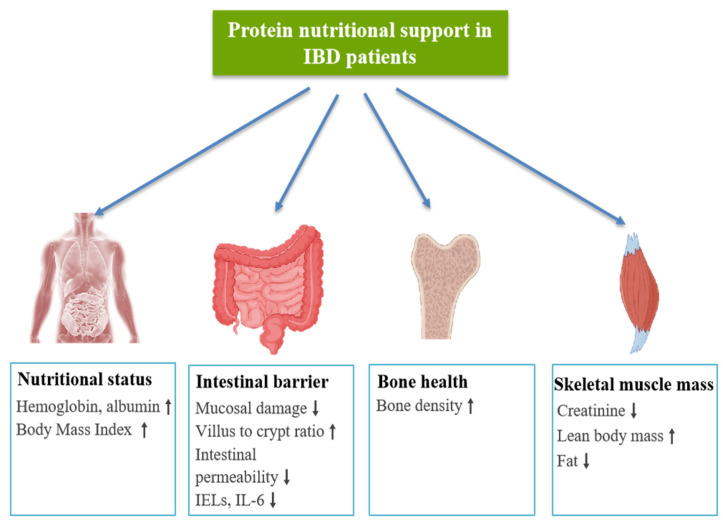
Protein nutritional support improves clinical indicators in IBD patients. ↑—increase, ↓—decrease.

**Table 1 nutrients-16-02302-t001:** The amelioration effect of high-quality animal and plant proteins on experimental IBD.

Dietary Protein Sources	Dosage	Model	Intervention Time	Conclusion	Reference
Soy protein	35% energy-providing protein intake	TNBS-induced colitis in SD rats	4 weeks	Soy protein improved podoplanin + infiltration of colonic mucosal structure; inhibited the proliferation of colon tumor necrosis factor-α + cells and RANKL expression; suppressed the expression of pro-inflammatory tumor necrosis factor-α and interleukin-6 in bone proteins; and mitigated the high osteoclast surface and depressed bone formation rate in TNBS rats.	[83]
Soy–Pea Protein	17.2% energy-providing protein intake	DSS-induced colitis in SAMP mice	6 weeks	Fecal myeloperoxidase (MPO) and FITC-glucan permeability scores were significantly decreased; the severity of cobblestone lesions decreased; the abundance of Lactobacillaceae and Leuconostraceae increased; and the concentration of metabolites glutamine and butyric acid increased, while the concentration of plasma linoleic acid decreased.	[28]
Soy protein	20% energy-providing protein intake	DSS-induced colitis in C57BL/6 mice	12 days	Soy protein reduced the content of mucin MUC1 and trefoil factor TFF-3 in the colon; inhibited the DSS-induced reduction in colon length; lowered the colon inflammation score; and reduced the expression of tumor necrosis factor-α in the colon and cecum.	[84]
Casein and whey protein concentrate	10% energy-providing casein + 10% energy-providing whey protein concentrate	DSS-induced colitis in BALB/c mice	3 weeks	Whey protein concentration can improve the loss of body mass in mice, and the effect of low-temperature treatment is more significant; low-temperature treatment of concentrated whey protein significantly reduced colonic inflammation and improved mucosal results; both low-temperature and high-temperature treatment of concentrated whey protein can increase the colonic mucin level; the level of myeloperoxidase in the colon of low-temperature whey protein concentration decreased; and low-temperature whey protein concentration down-regulated the expression of Gbp1, Gbp2, Gbp6 and Cxcl9.	[85]
Whey protein	18% energy-providing protein intake	DSS-induced colitis in Wistar rats	19 days	Whey protein reduced the expression of interleukin-1β, calprotectin, and inducible nitric oxide synthase; alleviated the clinical symptoms of diarrhea and fecal blood loss; increased the secretion of fecal mucin; and increased the expression of *Lactobacillus* and *Bifidobacterium*.	[86]
Whey protein	2.39 or 4.78 g/kg·body weight/day	Acetic acid-induced colitis in Wistar rats	7 days	Whey protein decreased the levels of inflammatory markers AP-1, COX-2, interleukin-6, interleukin-10, NF-κB, and tumor necrosis factor-α; up-regulated Nrf2 and HO-1 expression; and activated Nrf2/HO-1 pathway.	[87]
Acid casein and whey protein	14%, 30%, and 53% energy-providing protein intake (acid casein:whey protein = 5:1)	DSS-induced colitis in C57BL/6 mice	21 days	Compared with a dietary protein level of 14%, a 30% dietary protein diet increased epithelial repair by accelerating inflammation resolution, reducing colon permeability; 53% dietary protein diet aggravated DSS-induced inflammation.	[88]
Casein, whey protein, soy protein, white meat, and red meat	40% energy-providing protein intake	DSS-induced colitis in BALB/c mice	28 days	40% casein and red meat can exacerbate colitis; 40% whey protein can effectively alleviate colitis.	[26]
Casein or wheat gluten	20% and 60% energy-providing protein intake	DSS-induced colitis in BALB/c mice	35 days	The increase in animal protein resulted in a significant increase in colon Ly-6C^high^ monocytes and their activation; intestinal inflammation associated with anti-inflammatory TGF-β, pro-inflammatory cytokines (TNF-α and IL-1β), and inducible NO synthetase increased in mice fed a diet rich in animal protein, while plant-protein-rich diets generally decreased their expression.	[89]
Milk protein	53% energy-providing protein intake	DSS-induced colitis in C57BL/6 mice	14 days	Compared with the control group of animals that received DSS treatment, a high-protein diet is harmful in the later stages of induction, but it helps with the repair of colonic crypts after acute inflammation.	[90]
Whey proteins or donkey Whey proteins	0.2 g/d whey proteins or donkey whey proteins	DSS-induced colitis in C57BL/6 mice	21 days	Compared with the control group, both whey protein and donkey whey protein had the ability to inhibit the expression of proinflammatory protein and inflammatory secretion, and significantly decreased the levels of NF-κB and CD86; donkey whey protein is more effective than bovine whey protein in improving DSS induced colitis.	[91]
Dietary protein levels	14%, 30%, and 53% energy-providing protein intake	DSS-induced colitis in C57BL/6 mice	28 days	Compared with the other two diets, the diet with 30% protein content is associated with a lower protein synthesis rate, which can restore the initial level of the colon; it can restore colitis-induced changes such as body weight, cecal protein content, and spleen and muscle protein synthesis rates earlier; reduce inflammation and bacterial translocation in mice.	[92]
Whey protein hydrolysate	300 or 600 mg/kg·body weight/day	DSS-induced colitis in C57BL/6 mice	37 days	High doses of whey protein hydrolysate can significantly inhibit weight loss in mice with colitis, protect the colonic mucosal layer, significantly reduce the levels of inflammatory factors TNF-α, IL-6, and IL-1β in colitis mice; upregulate the secretion of short-chain fatty acids in colitis mice, and restore the imbalance of intestinal flora.	[27]
Alaska Pollock protein (APP)	20% energy-providing protein intake	DSS-induced colitis in C57BL/6 mice	51 days	APP intake inhibited DSS-induced weight loss, increased the disease activity index, increased spleen weight, shortened colon length, alleviated colonic tissue injury, and changed the structure and composition of fecal microbiota and short-chain fatty acids.	[93]
Quinoa protein or quinoa peptide	quinoa protein (1 g/kg·body weight/day) or quinoa peptide (500 or 1000 mg/kg·body weight/day)	DSS-induced colitis in C57BL/6 mice	35 days	Quinoa protein and quinoa peptide effectively relieve colitis symptoms: diarrhea, abdominal pain, bloody stool, weight loss, colon shortening, inflammatory factor release, and intestinal barrier damage. They also regulate gut microbiota, boost short-chain fatty acid production, and inhibit I-κB-α and NF-κB phosphorylation in colon tissues.	[94]

Abbreviations used: AP-1, active protein kinase-1; BALB/c, BALB/c mouse line; C57BL/6, C57BL/6 mouse line; CD86, cluster of differentiation 86; COX-2, cyclooxygenase-2; Cxcl,C-X-C motif chemokine ligand; DSS, dextran sulfate sodium; FITC, fluorescein isothiocyanate; Gbp, guanylate binding protein; HO-1, haem oxygenase-1; IL-1β, interleukin—1β; Ly-6C^high^, high expression of lymphocyte antigen 6 complex; MUC1, mucin 1; NF-κB, nuclear factor kappa-B; NO, nitric oxide; Nrf-2, nuclear-related factor-2; RANKL, receptor activator of nuclear factor-κ B ligand; SAMP, SAMP1/YitFC mouse line; SD, sprague–dawley; TFF-3, trefoil factors 3; TGF-β, transforming growth factor-β; TNBS, 2,4,6-trinitrobenzene sulfonic acid; TNF-α, tumor necrosis factor-α.

**Table 2 nutrients-16-02302-t002:** The application of protein nutritional intervention in the clinical treatment of patients with IBD.

Intervention Factors	Dosage	Age (Years)	Model	Intervention Time	Conclusion	Reference
Whey protein concentrate	0.5 g/kg·body weight/d	24~46	CD	8 weeks	Intestinal permeability and intestinal morphology were significantly improved; the villous crypt ratio was significantly increased. Inflammatory markers (intestinal epithelial lymphocytes, IELs) were reduced.	[30]
A diet rich in plant protein	-	26~41	Inactive CD	4 weeks	Significantly improved body composition in inactive and lactose intolerant CD patients; improved the treatment compliance of lactose intolerant patients.	[155]
Whey protein + resistance training (3 times weekly)	10 g/d whey protein + resistance training (3 times weekly)	30~58	IBD	8 weeks	Skeletal muscle mass was significantly higher than that of placebo group. Albumin, hemoglobin, and creatinine were significantly increased.	[31]
Whey protein + transforming growth factor(TGF)	-	16~62	CD	16 weeks	Lean body mass increased, while fat decreased.	[156]
Whey protein or soy protein isolate	30 g/d whey protein or 24 g/d soy protein isolate	-	CD	16 weeks	Both whey protein and soy protein isolate can reduce the triceps skinfold thickness and body fat percentage, while increasing the mid-arm circumference, correcting the arm muscle area, and increasing the percentage of lean body mass.	[29]
Hydrolyzed whey protein	Hydrolyzed whey protein (9.3–27.9 g/d)	-	CD	12 weeks	The improvement of nutritional status was related to the number of nutritional supplements; average albumin levels and body mass index improved; the index of disease activity decreased significantly; the average number of bowel movements per day decreased.	[32]
Dietary protein	20.7–152.5 g/d	27~45	CD	-	Protein intake was positively correlated with spinal bone density.	[157]
Casein glycomacropeptide	30 g/d	23~76	UC	8 weeks	A similar proportion of patients receiving casein glycomacropeptide or mesalamide dose escalation had a clinical response; colitis activity index and endoscopic lesion degree decreased; casein glycomacropeptide had good tolerance and acceptability.	[158]
Bovine lactoferrin	1 g/d	22	CD	42 months	After about 9 months, the disease activity index dropped from 50 to 35. After approximately 3 years and 5 months, blood tests showed no signs of disease relapse. The colonoscopy conducted at 3 years and 6 months revealed almost complete mucosal healing.	[159]
Lactoferrin	100 mg/d	9~14	IBD	3 months	Compared to ferrous sulfate, lactoferrin significantly increases hemoglobin, serum iron, and serum ferritin; lactoferrin significantly decreased interleukin 6 (IL-6) and hepcidin levels.	[160]

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
