# Peer review of "The Effect of Protein Nutritional Support on Inflammatory Bowel Disease and Its Potential Mechanisms"

_nutrients, 2024, doi:10.3390/nu16142302_

Round 1

Reviewer 1 Report

Comments and Suggestions for Authors

The manuscript "The effect of protein nutritional support on inflammatory bowel disease and its potential mechanisms" is an interesting and comprehensive review.

Title and  1. Introduction, 2. Pathogenesis, 3. Mechanisms of protein nutritional support in alleviating IBD has no problem.

In the section 4. The clinical application of protein nutritional support in IBD, ther is a question below; In CD treatment, elemental diet (amino-acid based diet) has been used for long term because of low antigenic stress and easy absorption. Please add discussion whey protein based treatment or elemental diet which is better for CD treatment.

5. Protein nutritional support on IBD complications,  6. Conclusions hasclear content; "Research findings reveal that protein nutritional support demonstrates significant benefits in improving clinical symptoms, reducing the risk of complications, and improving quality of life in IBD patients. Therefore, protein nutritional support is expected to provide a new approach for the treatment of IBD."

However, nutritional therapy is not incorporated into the guideliene yet.

 Many references are quoted and enoogh balance is shown.

Reviewer 2 Report

Comments and Suggestions for Authors

Li et al wrote a narrative review about the role of dietary proteins in the pathogenesis and management of inflammatory bowel disease. Main comments:

1) Page 2 lines 58-60: this sentence is quite vague, please be more precise.

2) Paragraphs 2.1, 2.2 and 2.3 are too long and too far from the aims of the review. Authors herein should focus on the role of dietary proteins.

3) Chapter 5 is too much theoretical. Authors should give practical indications for the clinicians about which is the ideal target intake for each situation and tips on how to achieve it.

4) What should be the attitude towards IBD patients under parenteral nutrition?

Round 2

Reviewer 2 Report

Comments and Suggestions for Authors

Answers were fine